# Fetal High-Density Lipoproteins: Current Knowledge on Particle Metabolism, Composition and Function in Health and Disease

**DOI:** 10.3390/biomedicines9040349

**Published:** 2021-03-30

**Authors:** Julia T. Stadler, Christian Wadsack, Gunther Marsche

**Affiliations:** 1Division of Pharmacology, Otto Loewi Research Center, Medical University of Graz, Universitätsplatz 4, 8010 Graz, Austria; 2Department of Obstetrics and Gynecology, Medical University of Graz, Auenbruggerplatz 14, 8036 Graz, Austria; christian.wadsack@medunigraz.at

**Keywords:** HDL, fetal development, pregnancy, sphingosine-1-phosphate, LpPLA_2_, gestational diabetes mellitus, preeclampsia

## Abstract

Cholesterol and other lipids carried by lipoproteins play an indispensable role in fetal development. Recent evidence suggests that maternally derived high-density lipoprotein (HDL) differs from fetal HDL with respect to its proteome, size, and function. Compared to the HDL of adults, fetal HDL is the major carrier of cholesterol and has a unique composition that implies other physiological functions. Fetal HDL is enriched in apolipoprotein E, which binds with high affinity to the low-density lipoprotein receptor. Thus, it appears that a primary function of fetal HDL is the transport of cholesterol to tissues as is accomplished by low-density lipoproteins in adults. The fetal HDL-associated bioactive sphingolipid sphingosine-1-phosphate shows strong vasoprotective effects at the fetoplacental vasculature. Moreover, lipoprotein-associated phospholipase A2 carried by fetal-HDL exerts anti-oxidative and athero-protective functions on the fetoplacental endothelium. Notably, the mass and activity of HDL-associated paraoxonase 1 are about 5-fold lower in the fetus, accompanied by an attenuation of anti-oxidative activity of fetal HDL. Cholesteryl ester transfer protein activity is reduced in fetal circulation despite similar amounts of the enzyme in maternal and fetal serum. This review summarizes the current knowledge on fetal HDL as a potential vasoprotective lipoprotein during fetal development. We also provide an overview of whether and how the protective functionalities of HDL are impaired in pregnancy-related syndromes such as pre-eclampsia or gestational diabetes mellitus.

## 1. Introduction

Cholesterol is an essential constituent in fetal development [1,2]. It has long been assumed that the fetus synthesizes most of its cholesterol requirements for growth de novo. However, in vitro and in vivo studies have shown that maternal circulating cholesterol can influence fetal metabolism [3,4]. Maternal dyslipidemia which is reflected either by an excess-, or also by limited cholesterol in the fetal circulation can affect fetal growth and health [5]. Maternally derived lipoproteins carrying cholesterol bind to their respective receptors expressed on the syncytiotrophoblast of the placental villi [6,7,8]. After cholesterol is taken up into the syncytium it is transported across the placental stroma to the fetal side. The exact transport mechanism is still elusive. At the endothelium of the fetoplacental vasculature, cholesterol is then transported via ATP-binding cassette G1 (ABCG1) or ATP-binding cassette A1 (ABCA1) to acceptors such as fetal HDL or lipid-poor apolipoproteins [4,9]. Interestingly, cholesterol is mainly carried by HDL in the fetal circulation, whereas in adults the majority of cholesterol is carried by low-density lipoproteins (LDL). Fetal HDL shows a unique composition and is suggested to exert different functions as in adults [10]. The current review focuses on the role of lipids carried by HDL in fetal circulation and the importance of exogenous cholesterol supply by the mother. Here, we summarize the proteomic composition of fetal HDL and highlight conspicuous changes compared to adult HDL. Further, we discuss the relationship of fetal HDL and sphingosine-1-phosphate (S1P) and the importance of S1P signaling at the fetoplacental vasculature in maintaining vascular integrity. Finally, the impact of pregnancy-associated disorders, such as preeclampsia (PE) and gestational diabetes mellitus (GDM) on HDL metabolism and function is comprehensively discussed.

## 2. Changes in Maternal Lipid Metabolism during a Normal Pregnancy

During pregnancy, multiple physiological changes occur that strongly influence maternal lipid metabolism. In the first two trimesters of pregnancy, maternal lipid metabolism is primarily anabolic and characterized by several factors that increase lipid accumulation in maternal tissues in preparation for the exponential increase in fetal energy requirements later in pregnancy [11,12]. These factors include maternal hyperphagia, to increase the availability of exogenous metabolic substrates [11,13] and an increase in insulin sensitivity which results in increased lipogenesis [14]. In the anabolic phase, hormonal and metabolic changes occur such as cortisol and leptin stimulation, and increased progesterone levels contribute to the accumulation of maternal fat depot [14].

During the last trimester of pregnancy, the lipid metabolism changes to the catabolic phase with a decline of fat accumulation [15]. This phase is characterized by increased lipolysis and mobilization of triglycerides from adipocytes. Furthermore, these changes are accompanied by a decrease of lipoprotein lipase (LPL) activity, leading to inefficient clearance of triglyceride-rich lipoproteins [16,17]. Maternal hyperlipidemia in late pregnancy coincides with changes in insulin sensitivity, which consistently decreases during this phase [18]. This decline is mediated by several factors, including increased levels of estrogen, placental lactogen, and progesterone [19].

During pregnancy, the lipid profile of mothers shows a 2.5-fold increase in very-low-density lipoprotein (VLDL) triglycerides and cholesterol and a 1.6-fold increase in LDL-cholesterol compared to non-pregnant women [20]. Plasma levels of VLDL and LDL steadily increase during gestation, while HDL levels show the highest rise in midgestation (45% above baseline) followed by a decline at term to about 15% [20]. HDL subclass analysis showed that levels of the triglyceride-rich HDL2 rise, while the smaller, lipid-poor HDL3 becomes less abundant [17]. These well-described alterations in lipoproteins, which are responsible for respective changes in maternal lipid profile during gestation are explained by several mechanisms: The increase of insulin resistance in late pregnancy mediates the elevated lipolytic activity in adipocytes, resulting in increased accessibility of substrates for triglyceride production in the liver [21,22]. Together with the decreased activity of LPL [17] and the stimulative effect of estrogen [23], these metabolic adaptions lead to an increased hepatic production of VLDL. The increased activity of the cholesteryl-ester transfer protein (CETP), which mediates the transfer of triglycerides on lipoproteins with higher density, contributes to the enrichment of triglycerides in HDL and LDL [17,24]. Another factor, contributing to the increase of triglyceride-rich HDL, is the reduced hepatic lipase activity, which reduces the clearance of HDL2 to smaller HDL3 [25].

Maternal hyperlipidemia during pregnancy is a prerequisite for delivering sufficient lipids of lipoproteins to the fetus. However, reduced or too high cholesterol supply to the fetus may lead to long-term consequences to the fetus [26].

## 3. Importance of Cholesterol in Fetal Development

Cholesterol is an essential constituent in embryonic and fetal development. It is a crucial component of cell membranes by defining fluidity and permeability. Further, cholesterol is an integral part of membrane microdomains, such as lipid rafts, which are essential for plasma-membrane-dependent signaling cascades. Cholesterol is a precursor of steroid hormones, including progesterone, and of its oxidative derivate oxysterol, which plays an important role in several metabolic processes [27].

The high requirements of cholesterol for the developing fetus have been described with 1.5–2.0 g of accumulated cholesterol per kg of added tissue [28]. The endogenous cholesterol originates from either de novo biosynthesis or hydrolysis of intracellular cholesteryl deposits by cholesterol esterases [29]. The fetus additionally possesses the capability to cover its demand of cholesterol from exogenous deposits. Yolk sac in early pregnancy and later the placenta has the same property to store maternally derived cholesterol [13]. The fact that the fetus does not rely on its own endogenously cholesterol was demonstrated in fetuses with the Smith-Lemli-Opitz syndrome, a condition with an inborn error of cholesterol synthesis. Fetuses affected by this syndrome harbor a nonsense mutation in the 7-dehydrocholesterol reductase, an enzyme that catalyzes the conversion of 7-dehydrocholesterol to cholesterol. Fetuses with this congenital condition are capable of developing to term, thereby demonstrating that maternal cholesterol needs to be transported across the placenta to maintain the demands of the fetus [13,30].

The human placenta is a unique organ, which is composed of several specialized cell types and mediates many metabolic exchange mechanisms between mother and fetus. To fulfill the demands of the fetus, nutrients and oxygen diffuse from maternal to fetal circulation by crossing directly into different cell layers. The first physical barrier, which limits nutrient transfer across the placenta is build up by the syncytiotrophoblast, a layer of multinucleated trophoblasts localized by the microvillous and basal membrane faced to the maternal and fetal side, respectively [31,32].

The first step of cholesterol transport from the mother to the fetus is the uptake on the apical, maternal side. Human placental trophoblasts express lipoprotein receptors such as scavenger receptor BI (SR-BI), LDL-receptor (LDL-R), and LDL receptor-related protein 1 (LRP1) (Figure 1) [4,6,7,8]. These receptors mediate the uptake of cholesterol and cholesteryl-esters from maternally derived lipoproteins [29,33]. After receptor-mediated endocytosis, the lipoprotein-associated cholesteryl-esters are intracellularly hydrolyzed [4]. Via Niemann-Pick C1 and/or other sparsely described cholesterol transporter proteins, free cholesterol is trafficked across the cell to membranes or metabolically active pools [4,34]. SR-BI mediates the selective uptake of cholesteryl-esters primarily from HDL, which are hydrolyzed by cytosolic cholesterol esterases and transported by potential carrier proteins to the basal membrane [4]. However, the exact pathway of transcellular cholesterol transport is still not known, but several transporters are thought to be involved, such as Niemann-Pick C1, Niemann-Pick C1-like protein 1, sterol carrier protein-x/2 and ABCA2. All these receptors are expressed in the human placenta [35]. To enter the fetal circulation, placental cholesterol needs to cross the endothelium at the fetoplacental vasculature. By using endothelial cells isolated from human term placentas, a study demonstrated efflux/secretion of exogenous cholesterol through ABCA1 and ABCG1 [9]. Acceptors of cholesterol in the cord blood are poorly lipidated apolipoprotein (apo) A-I (the major HDL associated apolipoprotein in adults), apoE, and HDL, with apoE-enriched HDL, was shown to be most efficient [9,35].

Maternally supplied cholesterol appears to be of great importance for fetal growth. Although there is no direct link between maternal and fetal lipoprotein metabolism, maternal serum cholesterol levels during pregnancy are directly related to infant birth weight. Low maternal serum cholesterol levels during pregnancy appear to increase the risk of microencephaly, while high maternal cholesterol levels promote the early incidence of atherogenicity [13,36]. Various further studies demonstrated a link between very high maternal cholesterol levels with prematurity and impaired fetal growth [37,38,39]. Dysregulated maternal cholesterol homeostasis during pregnancy has also been associated with disorders such as pregnancy-induced hypertension and preeclampsia [37,40,41].

Concluding, cholesterol plays an essential role in human fetal development and maternal hypocholesterolemia, as well as hypercholesterolemia, can affect fetal health and growth.

## 4. HDL Composition

HDLs are a group of highly heterogeneous lipoproteins, which are considered to have a high cardiovascular protective potential [42,43,44]. The heterogeneity of these particles depends on their size, shape, and compositional structure [45]. 

The major apolipoprotein in HDL is apoA-I, which accounts for around 70% of the total protein amount [46]. The second major apolipoprotein is apoA-II, which represents approximately 15–20% of total protein content [47]. The residual protein mass of HDL is composed of minor apolipoproteins, such as apoCs and apoA-IV, having an important enzyme regulatory function. ApoM is another crucial protein component on HDL, as it binds hydrophobic molecules, primarily sphingosine-1-phosphate (S1P) [48,49]. ApoE, apoD, apoF, apoJ, and apoL-I are further distinctly identified proteins on HDL whose exact roles have partly been identified. In addition, serum amyloid A (SAA), which is predominantly produced by the liver in the acute phase after an inflammatory stimulus, is mainly carried by HDL [50]. Furthermore, several enzymes are associated with HDL, including paraoxonase 1 (PON1), which has anti-inflammatory and antioxidative properties [51]. Direct binding of the enzyme to apoA-I on HDL stabilizes the protein and also stimulates PON1 lactonase activity [52]. Other HDL-associated enzymes are the lipoprotein-associated phospholipase A2 (LpPLA2) and lecithin-cholesterol-acyltransferase (LCAT). Additionally, enzymes with lipid transfer activity are important in HDL metabolism, including cholesterol ester transfer protein (CETP) and phospholipid transfer protein.

The most abundant lipids in HDL are phospholipids. Phospholipids and sphingolipids make up about 40–60% of the HDL lipidome, whereas cholesteryl-ester (30–40%), free cholesterol (5–10%), and triglycerides (5–12%) are not as prominent [53]. Like HDL-associated proteins, lipids of HDL also fulfill important structural functions. The ability of HDL to mediate cholesterol efflux is markedly modulated by the characteristics of its surface lipids. Therefore, phospholipids, which compose the surface lipid monolayer of HDL are an important determinant of its ability to accept cholesterol [53]. Moreover, both, the phospholipid content [53,54] and lysophospholipid content of HDL [55,56,57,58] markedly affect its anti-inflammatory properties. Sphingosine-1-phosphate (S1P) plays an important role in maintaining vascular homeostasis, which will be discussed in more detail in 7.2. Altogether, a total of 200 lipids and 80 proteins make up the diversity of different HDL subclasses [59,60,61].

## 5. HDL Functionality

### 5.1. Cholesterol Efflux Capacity

The best-studied property of HDL, which is also considered as the most clinically relevant atheroprotective function of HDL, is its ability to promote reverse cholesterol transport [62]. The uptake of excessive and accumulated cholesterol from peripheral cells is the first step of reverse cholesterol transport to the liver for catabolism. Given the heterogeneity of HDL particles in terms of structure and lipidomic/proteomic composition, steady-state HDL-cholesterol (HDL-C) levels suffer from the limitations inherent in their mass-based and static measurement. As a snapshot of the steady-state cholesterol pool, HDL-cholesterol levels do not provide direct information on the rate of cholesterol flux from vascular macrophages to the liver, which is influenced by many factors beyond the mass of HDL-C. Recent evidence clearly suggests that the cholesterol efflux capacity of HDL better reflects cardiovascular disease risk than HDL-C [63,64].

The reverse cholesterol transport starts with the release of lipid poor apoA-I from the liver and intestine, which circulates to peripheral cells to take up excess cholesterol, forming nascent HDL. ApoA-I is preferentially lipidated via ABCA1 [65], while cholesterol efflux to larger HDL subclasses is stimulated by ABCG1 [66,67]. Collectively, cholesterol can be actively transferred by SR-BI, ABCA1, and ABCG1, but also via passive diffusion [68,69,70]. After absorption from cells, cholesterol is esterified, catalyzed by LCAT, and large and mature HDL is formed. The HDL-associated cholesteryl-esters can be further transferred to LDL/VLDL by CETP. Thus, the transport of cholesterol from peripheral cells to the liver occurs via two pathways: Direct uptake by SR-BI and indirectly through HDL-LDL/VLDL interactions [71]. Reaching the liver, cholesteryl-esters are hydrolyzed and free cholesterol is either converted into bile acids, reused for the production of VLDL, or transferred by ABCG5/G8 into the bile.

### 5.2. Anti-Inflammatory and Antioxidative Capacities

Circulating HDL cholesterol concentrations do not provide information about the anti-inflammatory, antioxidant, antithrombotic, and endothelial function-promoting activities of HDL. In addition to its important role in reverse cholesterol transport, HDL can inhibit the transmigration of monocytes through endothelial and smooth muscle cell co-cultures [72]. HDL inhibits the expression of adhesion molecules, including vascular cell adhesion molecule, intercellular cell adhesion molecule, and E-selectin [73,74,75]. Through modulation of NF-κB and PPAR gamma, HDL further decreases the production of chemokines and chemokine receptors in vivo and in vitro [76]. Because of these properties, HDL diminishes the recruitment of monocytes, lymphocytes, and basophils to the vascular endothelium, thus slowing downstream processes of inflammatory response.

In addition to its numerous anti-inflammatory effects, HDL also possesses antioxidative properties. HDL protects LDL and other lipoproteins from oxidative damage induced by several oxidants, thereby reducing atherogenicity. ApoA-I plays a crucial role in the anti-oxidative capacity of HDL through the reduction of lipid hydroperoxides by their methionine residues [77,78]. The enzyme PON1 is associated with HDL and also contributes to the HDL-mediated antioxidative activity by reducing lipid peroxidation of LDL and HDL through a specific cysteine residue [79]. Other HDL-associated enzymes and apolipoprotein components, including LpPLA_2_, LCAT, apoA-II, apoE, and apoJ also contribute to HDL’s antioxidant properties [80,81,82]. Furthermore, HDL inhibits the formation of reactive oxygen species and reduces intracellular oxidative stress [83,84,85]. The attenuated cellular generation of ROS may be implicated in the antioxidative effect of HDL on endothelial cells [42,86].

### 5.3. Vasodilatory Activites

One of the most important functions of HDL is its vasodilatory effect, which is mainly seen in the increase in the availability of nitric oxide (NO) in the endothelial cells [87,88] and stimulating the generation and release of prostacyclin [89]. The initial step in the activation of NO production involves the binding of HDL to SR-BI, which initiates signaling in the endothelium [90]. The following intracellular events are facilitated by endothelial protein kinase B and intracellular Ca^2+^ mobilization, an increase in ceramide levels, and phosphorylation of endothelial NO synthase (eNOs) [42,87,91,92].

In addition, HDL, by its anti-oxidant activity, decreases the activity of nicotinamide adenine dinucleotide phosphate oxidase in the endothelium and decreases the formation of superoxide anions, which are potent inactivators of NO. Thereby, the bioavailability of NO is increased [93]. Vasodilatory actions of HDL also comprise the ABCG1 mediated efflux of cholesterol and 7-oxysterols, enhancing eNOs dimerization, leading to decreased production of reactive oxygen species [94].

## 6. Fetal Lipoproteins Show Altered Concentrations and Unique Composition

In cord blood, the concentration and composition of plasma lipoproteins are unique, suggesting that these particles may have an altered function in the developing fetus. While LDL represents the major class of lipoproteins in adult serum, HDL carries more than 50% of the cholesterol in fetal circulation. Although LDL and VLDL are detectable in the fetal circuit, but at low concentrations [95,96,97,98,99]. In the fetus, lipoproteins differ not only in concentrations but also in compositions, when compared with lipoproteins in adult plasma. In particular, the proteome of HDL has been shown to differ substantially from that in adults [10,99]. All studies investigating differences between maternal and fetal HDL found that only ApoE was present in higher concentrations compared to adult HDL, while all other apolipoproteins such as ApoA-I, ApoC-II, ApoC-III, and ApoD were lower (Figure 2) [10,95,100,101]. ApoA-I exerts a variety of important functions, such as interaction with cellular receptors, activation of LCAT, and anti-atherogenic activities [101,102,103]. ApoA-I further contributes to the anti-oxidative capacity of HDL, therefore lower levels in fetal HDL indicate diminished anti-oxidant function [10,95,101]. ApoE, which shows higher abundance on fetal HDL, plays an important role in cholesterol transport function by redistributing excess cholesterol from cells, to cells requiring it for metabolic processes such as membrane biosynthesis for cell proliferation or repair [104,105]. Large apoE enriched HDL particles are involved in the reverse cholesterol transport as ligands of SR-BI [9] and ABCG1 [105]. Further, apoE facilitates HDL binding to receptors of the LDL-receptor family [106]. In addition, apoE induces serum PON1 activity and stability comparable to apoA-I [107] and is reported as a major physiological activator of the lecithin-cholesterol acyltransferase (LCAT) [108]. Therefore, it appears that one function of HDL in the fetus is the transport of cholesterol to tissues as is accomplished by LDL in the adult [10,95]. Furthermore, studies on fetal HDL reported 5-fold lower PON1 mass and activity levels than in adults, which may be linked with a reduced anti-oxidative capacity and reduced defense against oxidative stress [10,109,110,111].

Although the proteomic differences between adult and cord blood HDL have been well described, there is currently no literature on the sphingolipid content of fetal HDL. The most abundant sphingolipid in HDL is sphingomyelin, which plays an important role in HDL functionality, by regulating fluidity and cholesterol efflux from different cells [112]. Furthermore, sphingomyelin affects the activity of enzymes involved in HDL metabolism and modulates the anti-oxidative properties of HDL [53,113]. Therefore, a highly interesting aspect for future studies would be to analyze the sphingolipidome of cord blood HDL, which could help to improve our understanding of the role and function of fetal HDL.

### HDL Metabolism in Cord Blood

Since there are strong differences in the fetal HDL composition, it is not surprising that HDL metabolism also significantly differs in fetal circulation. Sreckovic et al. showed that the activity of CETP was 55% lower in cord-compared to maternal serum, whereas LCAT activity did not differ [10]. Interestingly, it has been shown that CETP inhibition enhances the capacity of SR-BI and ABCG1 dependent efflux to the large HDL2 particles [114]. The decreased CETP activity and enrichment of HDL particles with apoE suggest a highly altered metabolism of HDL particles in fetal circulation [10]. Furthermore, analyses of subclass distribution revealed a shift in HDL subclasses, with a higher content of very large HDL particles, further supporting the hypothesis of a different physiological role of fetal HDL than in adults [109].

Fetal HDL is unique in every way, whether in composition or metabolism. However, there is not much literature on how these differences affect the function of HDL in the fetus and what specific physiological roles it may trigger.

## 7. The Role of Cord Blood-Derived HDL in Maintaining Fetoplacental Vascular Integrity

### 7.1. The Feto-Placental Endothelium

Understanding the mechanisms that underlie placental cholesterol transfer lies, at least in part, in the fetoplacental endothelium. The fetoplacental vasculature is unique in its lack of innervation, singular in being independent of the autonomic regulation to which other vascular beds are subject [115]. Therefore, locally produced vasoactive mediators such as NO, endothelin-1, and angiotensin II regulate placental vascular resistance [115,116,117]. Moreover, the placental vasculature responds differently to humoral factors than vessels in other vascular beds. For example, the placental vasculature is the only vascular bed that has been reported to constrict rather than dilate in response to prostaglandin E2. It also demonstrates blunted responses to other vascular mediators including acetylcholine, bradykinin, and angiotensin II [118,119,120]. Interestingly it has been shown that inhibited and impaired angiogenesis further contribute to placental vascular resistance in fetal growth-restricted pregnancies, creating structural changes that restrict blood flow [118]. This study underpins the importance of an adequate perfusion of the placental tissue for peri- and postnatal health of the offspring.

The fetoplacental circulation allows the villous arteries to carry deoxygenated and nutrient-depleted fetal blood via the cord from the fetus to the placenta. After the exchange of oxygen and nutrients in the tissue, the villous veins carry fresh oxygenated and nutrient-rich blood circulating back to the fetal systemic circulation [121].

Studies have shown that an imbalance in the production of these vasoactive agents in the placenta is associated with the incidence of pregnancy disorders [122,123].

### 7.2. HDL-Sphingosine-1-Phosphate (S1P) as an Important Regulator of the Feto-Placental Vasculature

S1P is a bioactive lipid and is involved in the regulation of the vasomotor tone through induction of NO and prostacyclin synthesis [87,124]. In the circulation, this sphingolipid is mainly produced by erythrocytes, platelets, and vascular endothelial cells [125,126]. Once released from these cells into the bloodstream, S1P mainly binds to HDL via binding to apoM, while a small fraction is transported by albumin or other lipoproteins [49]. It has been shown that the half-life of HDL-associated S1P is 4-fold increased, when compared to S1P linked to albumin, indicating the importance of the carrier protein [127]. S1P is a ligand for five different G protein-coupled receptors, named S1P receptors 1-5 (S1PR1-5) [128]. On endothelial cells, S1PR1-3 are expressed, with S1PR1 showing the highest abundance. Through interaction with S1PR1, S1P can activate several signal cascades, which play a key role in vascular homeostasis. Mice lacking the endothelial S1PR1 exhibit a pro-inflammatory phenotype, showing the significance of S1P-S1PR1 signal transduction on vascular protection [129]. Several studies suggest that S1P signaling is responsible for many of the cardio-protective properties of HDL, including the enhancement of endothelial barrier function and the induced vasodilator production [87,124,130]. Interestingly, during disorders such as cardiovascular disease or diabetes, the functionality of HDL-S1P has been shown to be reduced [131,132,133]. However, there are only a few available studies on the influence of HDL-S1P on the fetus and the fetoplacental unit.

In a study examining S1P in cord blood-derived HDL, S1P was shown to be present on fetal HDL and also bound to apoM, as is the case in the maternal circulation [134]. Further, S1PR1 was found as the predominant receptor expressed on the fetoplacental vasculature [134]. Ligation of S1P with its receptors elicits cell-type-specific cytoskeletal rearrangements [49,135,136]. Experiments on the effect of fetal HDL on cytoskeletal remodeling revealed that S1P-HDL isolated from cord blood triggers reorganization of actin filaments, resulting in an enhanced placental barrier function [134]. Using human umbilical vein endothelial cells, Wilkerson et al. also showed that HDL-associated S1P strengthens the endothelial barrier more persistently than albumin-bound S1P [137]. Moreover, Del Gaudio and colleagues observed that fetal HDL induces vasorelaxation of precontracted placental chorionic arteries [134]. The same authors further investigated the role of cord blood-derived HDL and S1P on the fetoplacental endothelium [138]. Primary fetal placental endothelial cells were approached by and challenged with TNFα to induce inflammation. They showed that incubation with fetal HDL-S1P complex from healthy donors diminished the ability of TNFα to activate signaling of NF-κB and expression of pro-inflammatory markers [138]. Angiotensin II is a stimulator of NADPH oxidase, which produces reactive oxygen species, leading to a vascular inflammatory response [139]. After treatment of primary fetal placental endothelial cells with angiotensin II, the production of reactive oxygen species was blunted in the presence of fetal HDL-S1P, whereas it was preserved when pre-incubated with an S1P receptor antagonist, suggesting that S1P signaling accounts for some of the vasculoprotective functions of HDL at the fetoplacental endothelium [138].

### 7.3. Protective Functions of Lipoprotein Associated Phospholipase A2 (LpPLA_2_) on the Feto-Placental Endothelium

The enzyme LpPLA_2_ is mainly produced by macrophages and binds to lipoproteins when secreted into circulation. In adults, LpPLA_2_ is mainly bound to LDL (80%), while the remainder is bound to HDL [140]. The preferred substrate for hydrolysis of LpPLA_2_ represents the platelet-activating factor (PAF), which is an important mediator of inflammation [141]. Activity and mass of LpPLA_2_ are altered in several pathologies such as hypercholesterolemia, diabetes, essential hypertension, and atherosclerosis and have therefore been the target of many clinical studies [142,143,144,145].

In a study focusing on LpPLA_2_ in the fetal circulation, HDL was identified as the major carrier, which is in contrast to adults [146]. In addition, this study reported that placental macrophages express LpPLA_2_, whose activity was increased by insulin, pro-inflammatory cytokines, and leptin [146]. Fetal HDL- LpPLA_2_ was shown to have a beneficial effect on endothelial barrier function, which was abrogated with a specific LpPLA_2_ inhibitor [146]. Interestingly, LpPLA_2_ levels in cord blood were inversely correlated with markers of oxidative stress [146]. These results suggest an important role of LpPLA_2_ on the placental endothelium and the fetus through athero-protective and anti-oxidative actions.

## 8. Pregnancy-Related Diseases Affects HDL Metabolism and Function

Severe changes in HDL metabolism as well as in parameters of HDL function have been reported in several inflammatory conditions including obesity [147,148,149], diabetes [150,151,152], cardiovascular disease [153,154], chronic kidney disease [155,156,157] or liver disease [158,159]. Impairment of HDL function may have pro-atherogenic properties and promote the inflammatory state. Changes in HDL functionalities have also been demonstrated in pregnancy-related diseases such as preeclampsia and gestational diabetes mellitus, which we will briefly summarize in the following chapter.

### 8.1. Preeclampsia Associated Changes in HDL Composition and Function

Preeclampsia (PE) is a hypertensive pregnancy-associated disorder, which develops usually after 20 weeks of gestation. This syndrome affects 2–8% of pregnancies worldwide and is a leading cause of maternal and fetal mortality [160,161]. This multiorgan disorder is defined as *de novo* hypertension (systolic blood pressure ≥ 140 mm Hg, diastolic blood pressure ≥ 90 mm Hg) and proteinuria (≥300 mg/24 h) [162]. Risk factors for the development of PE are pre-pregnancy body mass index, age, ethnicity (black women are at higher risk), primiparity, multiple pregnancies, and history of certain diseases before pregnancy such as chronic hypertension, diabetes mellitus, or renal disease [163]. In countries with low- and middle-income, PE and its convulsive form eclampsia account for 10–15% of direct maternal deaths [164,165]. This disorder is also associated with profound risks for the fetus including preterm birth, growth retardation, and death [166]. Mothers, affected by PE, but also their infants have a higher risk to develop cardiovascular disease later in life [167,168]. Nowadays, the only definitive treatment for PE is the management of clinical symptoms and delivery of the baby, which in turn increases the rate of preterm birth [165,169]. As the primary cause of PE, it has been suggested that impaired placentation and the subsequent systemic activation of the endothelium results in clinical manifestations [164].

During a healthy pregnancy, the vascular function has been shown to improve with gestational age [170], whereas obesity, a risk factor for PE, reduces endothelium-dependent and –independent vasodilation in mothers [171]. Interestingly, a study reported flow-induced dilatation in isolated vessels from healthy pregnant women, but not in arteries isolated from women diagnosed with PE [172]. These results suggest that enhanced responses to shear stress in the maternal circulation during pregnancy are important and, when absent as in PE, may contribute to the increase in maternal blood pressure [172].

Dyslipidemia in mothers diagnosed with PE has been reported in several studies, characterized by higher levels of total cholesterol, non-HDL-C, and triglycerides, but lower levels of HDL-C during the third trimester [173]. Due to the cardioprotective properties of HDL, changes in its function may contribute to the increased risk of cardiovascular events later in life in mothers, but also in children [174,175].

Einbinder et al. observed a decrease in PON1 lactonase activity in mothers affected by PE, indicating a decreased anti-oxidative and anti-inflammatory activity of HDL [176] (Figure 3). Moreover, they observed lower expression of endothelial NO synthase and an increased expression of the adhesion molecule VCAM-1 after preincubating human umbilical vein endothelial cells with isolated HDL from PE mothers [176]. Other studies focusing on structures of HDL and LDL in PE reported marked oxidative modifications, such as malondialdehyde and lipohydroperoxides in lipids and proteins of the isolated particles [177,178]. These results indicate that the markedly altered lipoprotein profile is due to PE-driven oxidative stress in the maternal systemic circulation. Other studies confirmed the reduction of PON1 activity in mothers suffering from PE, possibly due to PE-associated increased oxidative stress [177,179,180,181].

Of particular interest, the PE-associated oxidative modifications of lipids in HDL and LDL were also found in fetal lipoproteins, showing that also the infants are affected by increased oxidative stress and clear transplacental transmission of these effects in PE [182]. Similar to the results in PE mothers [177], PON1 activity was also shown to be decreased in the cord blood of the newborns [182]. In another study, HDL isolated from cord blood of PE pregnancies was reported to be linked with significantly reduced levels of apoM [138]. Given that levels of S1P are usually highly correlated with apoM levels, these results suggest less endothelial protection by this bioactive lipid [138]. Additionally, HDL from PE mothers has been shown to be depleted in apoM as well as S1P, accompanied with less anti-oxidative capacity [183].

Interestingly, another study reported an increased total- and HDL-mediated cholesterol efflux capacity of maternal and fetal PE sera, whereas ABCA1-mediated cholesterol efflux was decreased. This was partially explained by the increased concentration of apoE in maternal and fetal circulation. The authors proposed that the increased cholesterol efflux might be a rescue mechanism to remove excess cholesterol from cells to reduce lipid peroxidation [184].

Studies reported increased levels of LpPLA_2_ in maternal PE plasma and the placenta [185,186] as well as in the fetus [187], which may represent a compensatory mechanism to control PAF and inflammatory responses.

Alterations in HDL function and composition may contribute to the endothelial dysfunction observed in mothers affected by PE. Whether this impairment also contributes to the increased cardiovascular morbidity of these women and children later in life remains to be elucidated.

### 8.2. HDL in Gestational Diabetes Mellitus (GDM)

GDM is a condition in which women without a history of diabetes develop hyperglycemia during pregnancy. GDM is the most common disorder during pregnancy, affecting up to 22% of all pregnancies, with increasing prevalence worldwide [188,189]. Women diagnosed with GDM, have an increased risk of developing diabetes, hyperlipidemia, hypertension, and coronary heart disease later in life [190,191,192]. Therefore, lifelong health monitoring of these women is meanwhile recommended. Similar to PE, risk factors for GDM also include age, ethnicity, and obesity [193]. However, GDM not only affects the health of the mother but also fetal growth and the long-term health of the offspring. The most prominent adverse outcome of GDM complicated pregnancies represents macrosomia with complications including metabolic abnormalities, impaired immune system, degraded antioxidant status, and potential metabolic syndrome in adulthood [194].

In general, diabetes mellitus is associated with an altered lipid profile with increased levels of triglycerides, elevated LDL, and reduced levels of HDL [150]. Diabetic dyslipidemia is not only characterized by changed levels, but also by different structures, function, and metabolism of lipoproteins. Studies reported decreased levels of HDL in type 2 diabetes mellitus (T2DM), with predominance of the small, protein-rich particles, which can undergo rapid catabolism [150,195]. The changes in HDL subclass distribution result from the increased transfer of triglycerides on HDL, mediated by CETP [196] and the increased activity of lipolytic enzymes such as hepatic lipase [197,198,199]. There is increasing evidence that low HDL levels may have a direct impact on plasma glucose and thereby contribute to the pathophysiology of T2DM [200]. Several experimental and clinical studies have suggested that HDL lowers blood glucose levels, by increased uptake of glucose from skeletal muscle via activation of the AMP-activated kinase pathway [200,201,202] and further through stimulation of pancreatic β-cell insulin secretion [201,203,204]. Other properties of HDL, such as its pivotal role in reverse cholesterol transport, as well as its anti-inflammatory capabilities in immune cells and metabolic tissues, may contribute to enhanced insulin sensitivity [200].

In women with GDM, levels of triglycerides are markedly increased during pregnancy, while levels of HDL-C are decreased in the second and third trimesters [173]. Recent research on HDL subclass distribution in GDM revealed that small HDL particles are associated with GDM and provide a potential screening tool for early identification [205,206]. Mokkala et al. showed that women developing GDM have a distinct lipid profile in early pregnancy, with small-sized HDL particles being the strongest predictors for GDM [206].

In a study by Sreckovic et al., GDM-associated changes of HDL function and composition were examined in maternal as well as fetal HDL [207]. Shotgun proteomics of isolated HDL revealed lower levels of apoM and increased levels of the acute-phase reactant SAA on both, maternal and fetal GDM HDL [207] (Figure 4). Since apoM represents the main carrier of the vasoprotective S1P, the reduction of apoM on GDM HDL might contribute to endothelial dysfunction observed in GDM [208]. This was supported by another group using a migration assay with human umbilical vein endothelial cells [209]. Maternal GDM HDL showed less closure of cell migration, which was induced by TNF α, than control HDL [209]. Levels of apoA-I as well as mass and activity of PON1 were significantly decreased in maternal GDM HDL [207], similar to another study [209]. ApoA-I, as well as PON1, are important anti-oxidant components of HDL, therefore these results suggest decreased anti-oxidative protection [78,210]. On fetal GDM HDL, the abundance of PON1 was only barely detected, while activity was found to be reduced [207]. Interestingly, also HDL remodeling is altered during GDM. Both, maternal and fetal GDM HDL showed larger particle size than controls. Further, cholesterol efflux capacity was reduced in maternal as well as fetal GDM HDL [207]. Studies on LpPLA_2_ in GDM revealed higher activity on maternal as well as fetal HDL, which might be relevant to exert protective activities against oxidative stress [146].

Concluding, HDL proteome and size are markedly altered in GDM in both, maternal and fetal circulation. However, how these alterations affect the protective properties of fetal HDL and whether these alterations persist and are involved in the higher risk of becoming vascular diseases in offspring of GDM pregnancies later in life requires further studies.

## 9. Conclusions

Endogenous, as well as maternally-provided cholesterol, are important for fetal development. Although lipoprotein metabolism is separated between mother and fetus, maternal hyper- and hypocholesterolemia affect infant health and growth. Transplacental cholesterol transport from maternal lipoproteins to the fetal side involves receptor-mediated uptake of cholesterol from the syncytium and transport through the stroma. Cholesterol is then secreted/effluxed from the fetal endothelium to acceptors such as lipid-poor apolipoproteins and HDL.

With its unique apolipoprotein composition with high levels of apoE, fetal HDL seems to have an important cholesterol transport function that is accomplished by LDL in adults. Due to its distinct composition, it may also have an important role in atheroprotection. However, research should focus on elucidating the physiological function of fetal HDL and how this is developing with aging of the newborns.

Maintaining the vascular integrity of the fetoplacental vasculature is important for an adequate supply of oxygen and nutrients to the fetus and therefore crucial for fetal well-being. It has been shown that HDL-associated S1P is an important regulator of placental vascular inflammation, but also improves endothelial barrier function and induces vasorelaxation, thus playing an important role in maintaining vascular integrity. Further, LpPLA_2_ has been suggested to act anti-inflammatory and to improve vascular barrier function in the placental endothelium.

HDL composition and function have been shown to be altered in pregnancy disorders such as PE and GDM. Of importance, these changes were also observed in the fetus of complicated pregnancies, therefore suggesting placental transmission of these effects. Disease-induced alterations of HDL composition and function might contribute to the pathophysiology of PE and GDM. Long-term follow-up studies are needed to clarify whether alterations in HDL composition and function (i) persist into adulthood and (ii) whether these changes are related to the increased risk of vascular pathologies later in life.

This review summarizes the current literature on the composition and function of fetal HDL in health and disease. Extensive future research is needed to further understand the physiological role of HDL in the fetus.

## Figures and Tables

**Figure 1 biomedicines-09-00349-f001:**
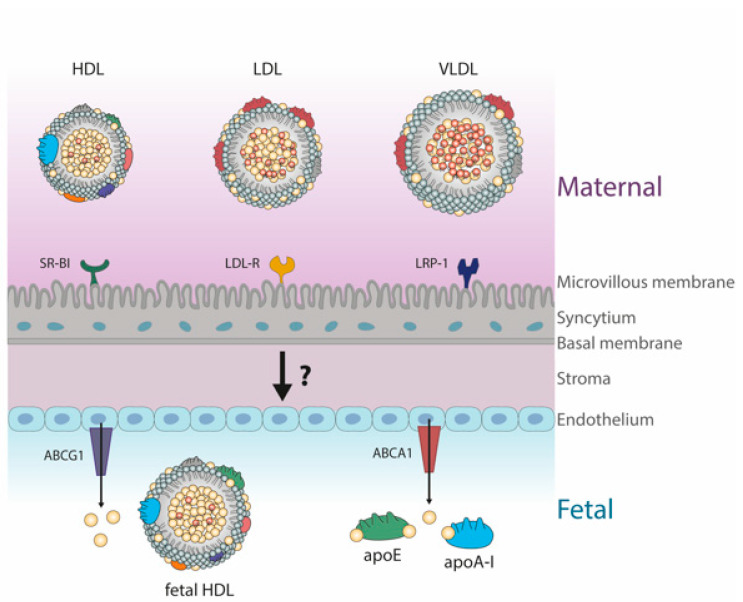
Described routes how maternal cholesterol is transported across the human placenta. First, maternally derived lipoproteins interact with respective receptors at the microvillous membrane of the syncytium. After uptake of cholesterol in the syncytium, it is secreted/effluxed to lipid-poor acceptor apolipoproteins of fetal HDL. How stroma transfers cholesterol to the fetoplacental endothelium remains elusive. High-density lipoprotein; SR-BI; scavenger receptor BI; LDL, low-density lipoprotein; VLDL, very-low-density lipoprotein; LDL-R, low-density lipoprotein receptor; LRP-1, LDL receptor-related protein 1; ABCA1, ATP-binding cassette A1; ABCG1, ATP-binding cassette G1.

**Figure 2 biomedicines-09-00349-f002:**
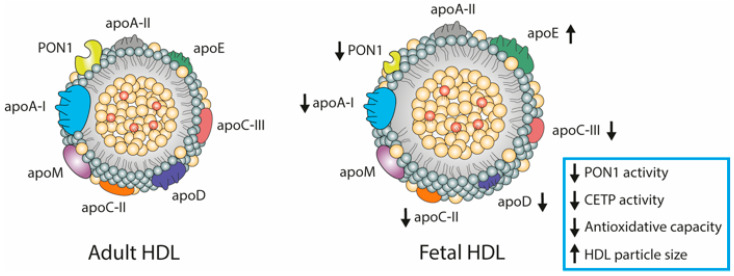
Schematic representation of differences between adult- and fetal HDL composition and function (indicated with black arrows). Cord blood-derived HDL exhibits several alterations in the apolipoprotein composition, such as decreased levels of apoA-I, apoC-III, apoD, and apoC-II and higher levels of apoE. In the fetus, the activity of CETP is decreased, while the mass and activity of PON1 and the antioxidative capacity are decreased. Fetal HDL is characterized by increased HDL particle size. HDL, high-density lipoprotein; apo, apolipoprotein; PON1, paraoxonase 1; CETP, cholesteryl-ester transfer protein.

**Figure 3 biomedicines-09-00349-f003:**
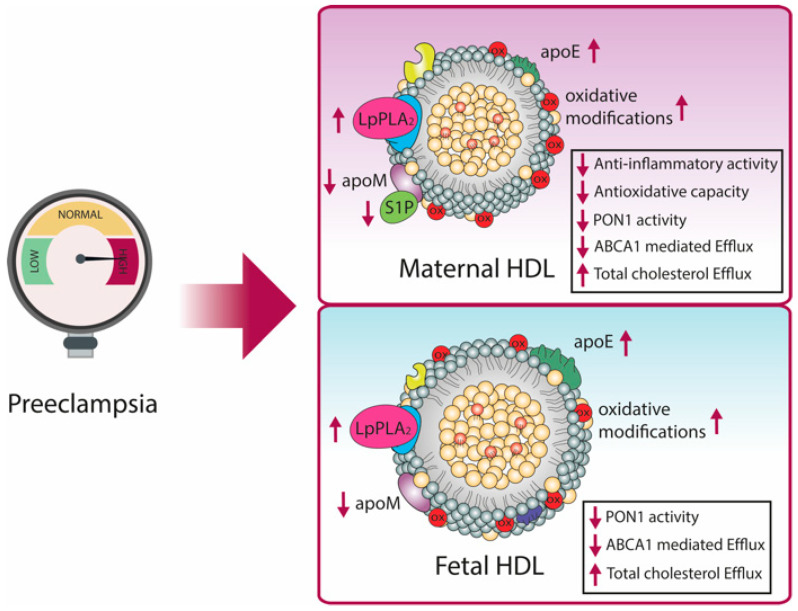
PE affects maternal and fetal HDL composition and function. Changes are indicated with purple arrows. In maternal HDL, a decrease in PON1 activity, apoM, and S1P content was observed, whereas apoE and LpPLA_2_ were increased. These changes in HDL composition were associated with reduced anti-inflammatory and anti-oxidative activity, but increased cholesterol efflux capacity. Fetal HDL of PE pregnancies showed similar changes, with reduced PON1 activity and apoM, but increased LpPLA_2_ and apoE, accompanied by increased cholesterol efflux capacity. Oxidative modifications of lipids were detected in both maternal and fetal HDL. HDL, high-density lipoprotein; apo, apolipoprotein; PON1, paraoxonase 1; LpPLA_2_, lipoprotein-associated phospholipase A2.

**Figure 4 biomedicines-09-00349-f004:**
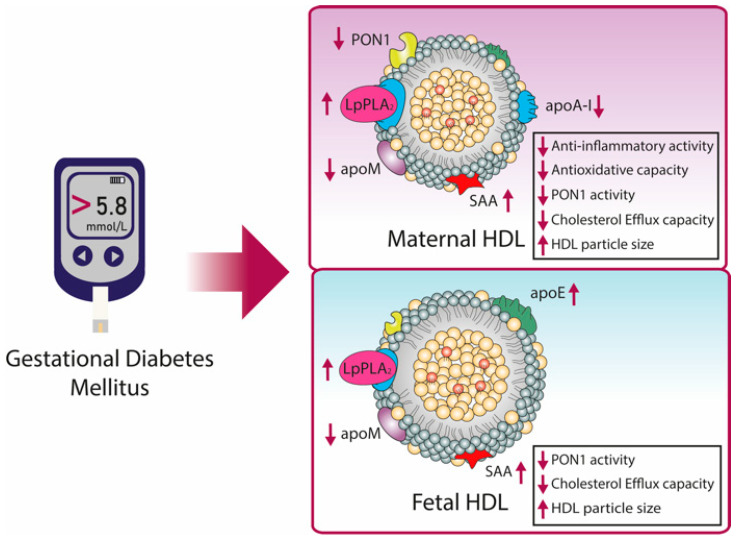
GDM affects maternal and fetal HDL composition and function. Changes are indicated with purple arrows. In maternal HDL, a decrease in PON1 activity and mass, apoM, and apoA-I content were observed, whereas SAA and LpPLA_2_ were increased. These changes in HDL composition were associated with reduced anti-inflammatory and anti-oxidative activity and reduced cholesterol efflux capacity. Fetal GDM-HDL showed the same alterations in PON1 and LpPLA_2_ activity, apoM and SAA content, and further increased apoE content. GDM was also accompanied by the increased particle size of both maternal and fetal HDL. HDL, high-density lipoprotein; apo, apolipoprotein; PON1, paraoxonase 1; LpPLA_2_, lipoprotein-associated phospholipase A2; SAA, serum amyloid A.

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
