# Peer review of "Fetal High-Density Lipoproteins: Current Knowledge on Particle Metabolism, Composition and Function in Health and Disease"

_biomedicines, 2021, doi:10.3390/biomedicines9040349_

Round 1

Reviewer 1 Report

The authors raise a very important clinical problem and relevant issue. 

Author Response

Reviewer: Thank you very much for allowing me to review manuscript entitled : “Fetal high-density lipoproteins: current knowledge on particle metabolism, composition and function in health and disease” submitted to „Biomedicines”.  . The purpose of this article was to assess the role of lipids carried by HDL in fetal circulation and the importance of exogenous cholesterol supply by the mother. The authors also discussed the relationship of fetal HDL and sphingosine-1-phosphate  and it's signaling importance at the feto placental vasculature in maintaining vascular integrity and the impact of pregnancy complications such as preeclampsia and gestational diabetes mellitus on HDL metabolism. It is a well done study. Citations and reference range are correctly selected. Despite the advances in medicine and many studies, the etiology of preeclampsia still remains not fully understood. Therefore, another voice in the discussion about etiopathogensis of this specific for human pregnancy complication such as preeclampsia and about the role fetal high-density lipoproteins in pregnancy complicated by preeclampsia is very valuable. Furthermore inclusion in the study of the assessment of changes in GDM is valuable and interesting. In this paper it has been suggested that HDL associated S1P is an important regulator of placental vascular inflammation, but also improves endothelial barrier function and induces vasorelaxation, thus playing an important role in maintaining vascular integrity. The Authors concluded that endogenous, as well as maternally provided cholesterol are important for fetal development. In my opinion, the Authors raise a very important topic and relevant issue . It is well done study. However, it seems that this is a preliminary study which  justifies undertaking further original experimental research in this field. It is a review paper and should constitute an introduction to further research in this area. The discussion is very interesting and the references are well chosen. Recommend this manuscript for publication in present version, maybe with an indication that this is a review based on the literature.

Response: We are pleased that our manuscript was favorably received by the reviewer and are happy to consider the helpful comments. According to the reviewer’s suggestion, we have added the following sentences to the conclusion of our manuscript (page 14): This review summarizes the current literature on the composition and function of fetal HDL in health and disease. Extensive further research is needed to further understand the physiological role of HDL in the fetus.

Reviewer 2 Report

Summary: The manuscript titled “Fetal high-density lipoproteins: current knowledge on particle metabolism, composition, and function in health and disease” is a comprehensive review focused on the role of lipids carried by HDL in fetal circulation and the importance of exogenous cholesterol supply by the mother. Authors include an in-depth overview of the most current knowledge on the proteomic and lipidomic composition of fetal HDL compared to adult maternal HDL. They also have some insight into the most common pregnancy-associated disorders, preeclampsia, and gestational diabetes mellitus, on HDL metabolism and function.

The review provides in-depth information about all the abovementioned areas with up-to-date, appropriate citations included. The manuscript is very well-written and easy to follow. The content is organized in a logical and easy-to-read manner. Authors provide an appropriate well-balanced mix of critical analysis of existing information in the field and self-formed comments and relevant and informative conclusions. Each section of the manuscript is well-defined and contains only information relevant to it. Figures included in the document are of excellent quality, easy to read and understand.

A minor suggestion to the authors is, if possible, to expand a little their review of the sphingolipidome (ceramide, sphingomyelin, and separate species) of maternal vs. fetal HDL and potential changes in the diseases (data availability permitting).

Author Response

Summary: The manuscript titled “Fetal high-density lipoproteins: current knowledge on particle metabolism, composition, and function in health and disease” is a comprehensive review focused on the role of lipids carried by HDL in fetal circulation and the importance of exogenous cholesterol supply by the mother. Authors include an in-depth overview of the most current knowledge on the proteomic and lipidomic composition of fetal HDL compared to adult maternal HDL. They also have some insight into the most common pregnancy-associated disorders, preeclampsia, and gestational diabetes mellitus, on HDL metabolism and function.

The review provides in-depth information about all the abovementioned areas with up-to-date, appropriate citations included. The manuscript is very well-written and easy to follow. The content is organized in a logical and easy-to-read manner. Authors provide an appropriate well-balanced mix of critical analysis of existing information in the field and self-formed comments and relevant and informative conclusions. Each section of the manuscript is well-defined and contains only information relevant to it. Figures included in the document are of excellent quality, easy to read and understand.

A minor suggestion to the authors is, if possible, to expand a little their review of the sphingolipidome (ceramide, sphingomyelin, and separate species) of maternal vs. fetal HDL and potential changes in the diseases (data availability permitting).

Response: 

We are glad to hear that our manuscript was positively received by the reviewer and we are happy to consider the helpful comment.

Unfortunately, there is currently no literature on the sphingolipid content of fetal HDL.

However, we have included the following part in our review (page 7). "Although the proteomic differences between adult and cord blood HDL have been well described, there is currently no literature on sphingolipid content of fetal HDL. The most abundant sphingolipid in HDL is sphingomyelin, which plays an important role in HDL functionality, by regulating fluidity and cholesterol efflux from different cells [113]. Furthermore, sphingomyelin affects the activity of enzymes involved in HDL metabolism and modulates anti-oxidative properties of HDL [53,114]. Therefore, a highly interesting aspect for future studies would be to analyze the sphingolipidome of cord blood HDL, which could help to improve our understanding of the role and function of fetal HDL"